# An index for quantum integrability

**Shota Komatsu[1*], Raghu Mahajan[1,2†] and Shu-Heng Shao[1‡]**

**1** School of Natural Sciences, Institute for Advanced Study, Princeton, NJ 08540, USA
**2** Department of Physics, Princeton University, Princeton, NJ 08540, USA

★ skomatsu@ias.edu † raghu_m@princeton.edu ‡ shao@ias.edu

## Abstract

The existence of higher-spin quantum conserved currents in two dimensions guarantees quantum integrability. We revisit the question [1] of whether classically-conserved local higher-spin currents in two-dimensional sigma models survive quantization. We define an integrability index $\mathfrak{I}(J)$ for each spin $J$, with the property that $\mathfrak{I}(J)$ is a lower bound on the number of quantum conserved currents of spin $J$. In particular, a positive value for the index establishes the existence of quantum conserved currents. For a general coset model, with or without extra discrete symmetries, we derive an explicit formula for a generating function that encodes the indices for all spins. We apply our techniques to the $\mathbb{CP}^{N-1}$ model, the $O(N)$ model, and the flag sigma model $\frac{U(N)}{U(1)^N}$. For the $O(N)$ model, we establish the existence of a spin-6 quantum conserved current, in addition to the well-known spin-4 current. The indices for the $\mathbb{CP}^{N-1}$ model for $N > 2$ are all non-positive, consistent with the fact that these models are not integrable. The indices for the flag sigma model $\frac{U(N)}{U(1)^N}$ for $N > 2$ are all negative. Thus, it is unlikely that the flag sigma models are integrable.


## 1 Introduction

Starting with the seminal work of [2] and [3], it has been known that there exist integrable quantum field theories in two dimensions whose S-matrices factorize. This property is tied to the existence of higher-spin conserved currents [4]. More precisely, it was shown in [5] that the existence of *one* local higher-spin current is sufficient for factorization of the S-matrix in parity symmetric theories, while one needs two currents in theories without parity.

However, even for sigma models with a coset target space, a complete understanding and classification of quantum conserved currents is still lacking. See, for example, [6,7] for reviews on integrability in two-dimensional sigma models. At the classical level, it is known that sigma models whose target space is a symmetric coset admit a so-called Lax operator formalism, which allows one to systematically construct classically-conserved local higher-spin currents [8]. However, the coset being symmetric is neither sufficient nor necessary to diagnose the fate of classical integrability at the quantum level. It is insufficient because in some symmetric coset sigma models, higher-spin currents fail to be conserved at the quantum level. A famous example is the $\mathbb{CP}^{N-1}$ sigma model [1]. It is not necessary either, since, even if the coset is not symmetric, one can sometimes construct a Lax operator. Interesting examples are sigma models on the Schrödinger spacetime [9–11].

An approach to directly address quantum integrability was presented by Goldschmidt and Witten [1], where they provided a sufficient condition for the existence of quantum conserved currents in two-dimensional sigma models.[1] Their analysis, which we review below, is based on the fact that any sigma model, be it a symmetric coset or not, is conformal at the classical level and has a current for every even integer spin $2n$ built from the stress tensor:

$$(\mathcal{J}_+^{\text{cl}}, \mathcal{J}_-^{\text{cl}}) := ((T_{++})^n, 0). \tag{1.1}$$

Owing to the fact that $\partial_- T_{++} = 0$, this current is conserved classically

$$\partial_- (T_{++})^n = 0 \quad \text{(classical)}. \tag{1.2}$$

---

[1] In this paper we will only consider quantum charges built from local conserved currents. For non-local quantum charges, see, for example, [12–14]. In particular, it was shown that a sufficient (but not necessary) condition for the conservation of the non-local charges in $G/H$ coset sigma models is that $H$ is simple [14]. Integrable examples with $H$ not simple include $O(2N)/O(N) \times O(N)$ [15,16] and $Sp(2N)/Sp(N) \times Sp(N)$ [17], where the quantum conservation of the non-local charges is secured by a $\mathbb{Z}_2$ discrete symmetry [18]. A similar analysis was performed for the superstring sigma model on $AdS_5 \times S^5$ in [19]. The relation between the local and non-local charges was discussed in [20]. See also [21] for discussions on the relation between the non-local charges and the factorization of the S-matrices.

At the quantum level, the conservation law of this higher-spin current is generally broken and the classical equation (1.2) is modified to

$$\partial_- (T_{++})^n = A \quad \text{(quantum)}, \tag{1.3}$$

where $A$ is some local operator with classical dimension $2n + 1$ and spin $2n - 1$. This is the standard way in which classical integrability fails to generalize to quantum integrability.

However, if $A$ can be written as a total derivative, i.e. if there exist operators $B_+$ and $B_-$ so that $A = \partial_+ B_- + \partial_- B_+$, one can redefine the current such that it is still conserved. Explicitly,

$$(\mathcal{J}_+^{\text{qu}}, \mathcal{J}_-^{\text{qu}}) := ((T_{++})^n - B_+ \,, \, -B_-) \,, \tag{1.4}$$

$$\partial_- \mathcal{J}_+^{\text{qu}} + \partial_+ \mathcal{J}_-^{\text{qu}} = 0 \,. \tag{1.5}$$

Thus, there is still a conserved current $(\mathcal{J}_+^{\text{qu}}, \mathcal{J}_-^{\text{qu}})$ at the quantum level. In a given theory, one can in principle identify all possible $A$ terms, and total derivative terms $B$, that have the right quantum numbers to appear on the right hand side of (1.3). The message of [1] is that if the number of $A$ terms is less than or equal to the number of $B$ terms, it is guaranteed that there are conserved higher-spin currents at the quantum level, because any correction on the RHS of (1.3) can be written as a total derivative, and thus absorbed into a redefinition of the current.

More generally, there might be more than one classically-conserved current of a given spin, rather than just $(T_{++})^n$ [22, 23]. This motivates us to consider the combination

$$\mathcal{I}(J) :=$$
$$(\# \text{ of classically-conserved spin-}J \text{ non-derivative currents}) - [(\# \text{ of } A\text{'s}) - (\# \text{ of } B\text{'s})] \,. \tag{1.6}$$

From the first term on the right hand side, we have to omit currents that are derivatives (e.g. $\partial_+ T_{++}$) because they do not give rise to a charge when integrated on a spatial slice. If $\mathcal{I}(J) > 0$, it is guaranteed that there are $\mathcal{I}(J)$ quantum conserved currents by the argument of [1] that we reviewed above. Hence a positive $\mathcal{I}(J)$ for some $J > 2$ provides a *sufficient* condition for quantum integrability. It is not a necessary condition because it is possible that even though $\mathcal{I}(J) \leqslant 0$ in some case, the model might be fine-tuned such that quantum conserved currents still exist.

In [1], Goldschmidt and Witten considered the classically-conserved current $(T_{++})^2$, and enlisted the possible $A$ and $B$ operators by brute force in some specific examples. However, the complexity of this brute-force method quickly goes out of control for larger spin, as well as for a more general target space. In this paper we systematize the computation of [1] to general coset models $G/H$. Our analysis is based on a simple observation: The difference $\mathcal{I}(J)$ is invariant under conformal perturbation theory around the UV fixed point, while individual numbers in (1.6) are not.[2] This allows us to compute $\mathcal{I}(J)$ at the UV fixed point where the equation of motion simplifies and the theory enjoys conformal symmetry. Because of this property, we call $\mathcal{I}(J)$ the *integrability index*. A similar quantity been studied by Zamolodchikov [24] in the context of integrable perturbation of conformal field theory.

We derive a compact expression for a generating function that allows us to compute the indices $\mathcal{I}(J)$ for all spins. Our technique is largely inspired by [25] which classified higher-dimension operators in effective field theory, and also by the computation of supersymmetric indices [26]. As one important application, we compute the indices for the $O(N)$ model and establish the existence of a spin-6 quantum conserved current in addition to a spin-4 current predicted in [1].

---

[2]See however the discussion at the end of subsection 3.4 for potential subtleties due to nonperturbative corrections.

The coset model $G/H$ typically has some tunable parameters. Over certain loci in this parameter space, there could be extra discrete global symmetries, and these can affect quantum integrability. The importance of the discrete symmetries for quantum conservation of non-local and local charges was emphasized in [18]. The classic example is the $\mathbb{CP}^1$ sigma model, which has a $2\pi$-periodic $\theta$ angle. The model is integrable at $\theta = 0$ [2] and at $\theta = \pi$ [27] where there is an extra $\mathbb{Z}_2$ charge conjugation symmetry. At other values of $\theta$, there is no extra symmetry and the model is not expected to be integrable. Thus, we will also present a generating function and compute $\mathfrak{I}(J)$ in the presence of discrete global symmetries. For example in the $\mathbb{CP}^1$ model, we find that $\mathfrak{I}(4) \leqslant 0$ and $\mathfrak{I}(6) \leqslant 0$ without imposing the $\mathbb{Z}_2$ charge conjugation symmetry, while $\mathfrak{I}(4) = \mathfrak{I}(6) = +1$ when the $\mathbb{Z}_2$ symmetry imposed. This is consistent with quantum integrability at $\theta = 0$ and $\theta = \pi$. For the $\mathbb{CP}^{N-1}$ models with $N > 2$, the indices are all negative even after including the discrete symmetry, consistent with the standard lore that the $\mathbb{CP}^{N-1}$ model for $N > 2$ is not integrable.

Finally, we apply our formalism to the flag sigma models $\frac{U(N)}{U(1)^N}$, which reduces to the $\mathbb{CP}^1$ model when $N = 2$.[3] Aspects of flag sigma models, including their global symmetries, 't Hooft anomalies and phase diagrams have recently received some attention [28–34]. In particular, it was argued that over certain loci in parameter space with enhanced discrete global symmetry, the IR phase is gapless and is described by the $SU(N)_1$ WZW model [30–32]. We compute the indices $\mathfrak{I}(J)$ for these models, and we find that they are all negative. Thus, it is unlikely that these models are integrable.

The organization of the paper is as follows. In Section 2 we review the Lagrangian description of coset sigma models and present a complete set of "letters" for constructing local operators. In Section 3 we construct the partition function using a plethystic exponential, define the integrability index and discuss the invariance of the index in conformal perturbation theory. In Section 4, we work out the partition function and the index for the $\mathbb{CP}^{N-1}$, $O(N)$ and the flag sigma models. We conclude in Section 5 and discuss directions for future work.

## 2 Coset sigma models

### 2.1 Lagrangian description of coset models

Let us first review the basic properties of sigma models with a coset target space $G/H$. We do *not* require the coset to be symmetric. Let $\mathfrak{g}$ and $\mathfrak{h}$ be the Lie algebras of $G$ and $H$, respectively. Using the quadratic form $\langle \, , \rangle$ on $\mathfrak{g}$, we can make an orthogonal decomposition of $\mathfrak{g}$ as

$$\mathfrak{g} = \mathfrak{h} \oplus \mathfrak{k}. \tag{2.1}$$

The elements in $\mathfrak{k}$ represent the physical degrees of freedom of the coset. To keep a concrete example in mind, consider the coset $\frac{SU(2)}{U(1)}$, which is nothing but the $O(3)$ or the $S^2$ or the $\mathbb{CP}^1$ model. In this case, the full Lie algebra $\mathfrak{g} = \mathfrak{su}(2)$ is spanned by the three Pauli matrices, $\mathfrak{h}$ is the span of the Pauli-Z matrix, and $\mathfrak{k}$ is the span of the Pauli-X and Pauli-Y matrices.

We will work on two-dimensional Minkowski space $\mathbb{R}^{1,1}$ throughout. The target space of the sigma model is the space $G/H$ of all *left* cosets of $H$. To write the action, we first consider all maps $g : \mathbb{R}^{1,1} \to G$ and then proceed to impose the following local symmetry:

$$g(x) \to g(x) h(x)^{-1}, \quad h(x) \in H. \tag{2.2}$$

---

[3]The $\mathbb{CP}^1$ model has another generalization $SU(N)/SO(N)$ with global symmetry $PSU(N)$. This model has a $\mathbb{Z}_2$-valued $\theta$-angle for $N > 2$. For both values of $\theta$, the model is integrable. At $\theta = 0$, the IR phase is gapped, while at $\theta = \pi$ the IR is described by the $SU(N)_1$ WZW model [15, 16].

In other words, we have to make the identification of maps $g(x) \sim g(x)h(x)^{-1}$ for any $h : \mathbb{R}^{1,1} \to H$. This restricts us to maps from the spacetime into the space $G/H$ of left-cosets of $H$. This model admits a *global $G$-symmetry*[4] which acts from the left (contrasted with the local $H$ symmetry which acts from the right):

$$g(x) \to g'g(x), \quad g' \in G.\tag{2.3}$$

To write down the action of the sigma model, we introduce the left-invariant one-form $j$

$$j_\mu(x) := g^{-1}(x)\partial_\mu g(x).\tag{2.4}$$

Since $j$ is valued in the Lie algebra, one can decompose it using (2.1) as

$$j_\mu(x) = a_\mu(x) + k_\mu(x), \qquad a_\mu(x) \in \mathfrak{h}, \quad k_\mu(x) \in \mathfrak{k}.\tag{2.5}$$

The currents $j_\mu(x)$, $a_\mu(x)$ and $k_\mu(x)$ are invariant under the global $G$ transformations (2.3), while they transform under the local $H$ transformations (2.2) as

$$j \to h\,j\,h^{-1} - dh\,h^{-1},\tag{2.6}$$

$$a \to h\,a\,h^{-1} - dh\,h^{-1},\tag{2.7}$$

$$k \to h\,k\,h^{-1}.\tag{2.8}$$

In particular, $a_\mu(x)$ transforms as a gauge field under the local action of $H$ from the right. The covariant derivative built out of $a_\mu$

$$D_\mu := \partial_\mu + a_\mu\tag{2.9}$$

transforms via conjugation under the $H$ gauge transformations, $D_\mu \bullet \to h\,(D_\mu \bullet)\,h^{-1}$.

The action for the sigma model with target space $G$ (without topological terms) is $\int d^2x \operatorname{tr} j_\mu j^\mu$. Now we have to gauge the $H$ symmetry. For that purpose, we introduce a gauge field $A_\mu \in \mathfrak{h}$ and the covariant derivative acting on $g(x)$ as $g^{-1}\mathbb{D}_\mu g = g^{-1}\partial_\mu g - A_\mu$. Now we can manipulate the action

$$
\begin{aligned}
\operatorname{tr}(g^{-1}\mathbb{D}_\mu g)^2 &= \operatorname{tr}(g^{-1}\partial_\mu g)^2 + \operatorname{tr}A_\mu^2 - 2\operatorname{tr}A_\mu(g^{-1}\partial_\mu g) \\
&= \operatorname{tr}(k_\mu^2 + a_\mu^2) + \operatorname{tr}A_\mu^2 - 2\operatorname{tr}A_\mu a_\mu \\
&= \operatorname{tr}k_\mu^2 + \operatorname{tr}(a_\mu - A_\mu)^2.
\end{aligned}
$$

In going to the second line, we split $g^{-1}\partial_\mu g = j_\mu = a_\mu + k_\mu$ and used the orthogonality of $\mathfrak{h}$ and $\mathfrak{k}$. Integrating out $A$, we see that the action of the sigma model can be expressed as

$$S[g] = \frac{R^2}{2} \int d^2x\ \operatorname{tr}\left[k_\mu(x)\,k^\mu(x)\right],\tag{2.10}$$

where the positive real number $R$ characterizes the size of the coset. As desired, the action is invariant under the local $H$ transformation of $k$ (2.8). We have used the notation $S[g]$ on the left hand side to emphasize the fact that we start with maps $g : \mathbb{R}^{1,1} \to G$ and then view $a_\mu(x)$ and $k_\mu(x)$ as being determined by $g(x)$. Even though it might seem that there is no $a_\mu(x)$ dependence on the right hand side, this is not the case, as will become explicit in the equation of motion (2.13) below.

---

[4]To be precise, the global symmetry may not be $G$, but a discrete quotient thereof. For example, the global symmetry of $\mathbb{CP}^{N-1} = \frac{SU(N)}{U(N-1)}$ is $PSU(N)$ and not $SU(N)$. This does not affect our arguments in this section, but the role of discrete symmetry will become important later.

Let us now derive the equation of motion starting from the action (2.10). We make the first order variation $g \to (1 + \epsilon)g$, and write the variation of the Lagrangian from (2.10) as being proportional to $k^\mu \delta k_\mu$. Under $g \to (1 + \epsilon)g$, the variation of the current $j_\mu$ is $\delta j_\mu = g^{-1}(\partial_\mu \epsilon)g$. Now we use $\delta k_\mu = \delta j_\mu - \delta a_\mu = g^{-1}(\partial_\mu \epsilon)g - a_\mu$, and the orthogonality of $\mathfrak{h}$ and $\mathfrak{k}$ to get

$$\delta S = R^2 \int d^2 x \, \mathrm{tr} \left[ \partial_\mu \epsilon \, (g \, k^\mu \, g^{-1}) \right]. \tag{2.11}$$

Therefore, the equation of motion reads

$$\partial_\mu \left( g \, k^\mu \, g^{-1} \right) = 0. \tag{2.12}$$

This is equivalent to $\partial_\mu k^\mu + [j_\mu, k^\mu] = 0$. We now make the decomposition $j_\mu = a_\mu + k_\mu$ as in (2.5) and since $[k_\mu, k^\mu] = 0$, we get an equivalent form of the equation of motion

$$D_\mu k^\mu(x) = \partial_\mu k^\mu(x) + [a_\mu(x), k^\mu(x)] = 0, \tag{2.13}$$

where the covariant derivative $D_\mu$ acting on adjoint fields was defined in (2.9). The equation of motion (2.12) also shows that the current

$$J^\mu(x) := g(x) \, k^\mu(x) \, g^{-1}(x) \tag{2.14}$$

is conserved $\partial_\mu J^\mu(x) = 0$. From the variation of the action (2.11), we see that $J^\mu(x)$ is nothing but the Noether current of the global $G$ symmetry. Let us also comment that $J_\mu(x)$ is invariant under the local $H$ transformations (2.2) and (2.8).

We end this section with a result that we shall use later. The identity $d(g^{-1}dg) + g^{-1}dg \wedge g^{-1}dg = 0$ implies that $\partial_\mu j_\nu - \partial_\nu j_\mu + [j_\mu, j_\nu] = 0$. Writing $j = a + k$ and decomposing this identity into $\mathfrak{h}$ and $\mathfrak{k}$ sectors, we get

$$[D_\mu, D_\nu] = -[k_\mu, k_\nu]\Big|_{\mathfrak{h}}, \tag{2.15}$$

$$D_\mu k_\nu - D_\nu k_\mu = -[k_\mu, k_\nu]\Big|_{\mathfrak{k}}, \tag{2.16}$$

where the right-hand-sides designate the restriction of $[k_\mu, k_\nu]$ to $\mathfrak{h}$ or $\mathfrak{k}$. For a symmetric coset, $[k_\mu, k_\nu]|_{\mathfrak{k}} = 0$.

## 2.2 Description of local operators

We are interested in classifying the possible $A$ terms that appear in the conservation law of a conserved current of spin $J$. For concreteness, we only consider the case when the current is a singlet under the global $G$ symmetry, as this is relevant for operators like $(T_{++})^n$. Our methods can be adapted to the case when the current transforms nontrivially under the global $G$ symmetry.

We need a way to count local operators that are invariant both under the global $G$ symmetry and the $H$ gauge transformations. Such analysis was performed in [25] for effective field theories in higher dimensions, and we apply their techniques to coset models in two dimensions. Local operators can be built from $g(x)$, $k_\mu(x)$, and their covariant derivatives. Let us recall the transformation properties of these fields

$$g(x) \to g' \, g(x) \, h^{-1}(x), \tag{2.17}$$

$$k_\mu(x) \to h(x) \, k_\mu(x) \, h^{-1}(x), \tag{2.18}$$

$$D_\mu \bullet \to h(x) \, (D_\mu \bullet) \, h^{-1}(x). \tag{2.19}$$

First, we show that the fields $g(x)$ and $g^{-1}(x)$ can be omitted from this list. To see this, note that the covariant derivatives acting on $g(x)$ or $g^{-1}(x)$ can be written as

$$D_\mu g = \partial_\mu g - g a_\mu = g k_\mu, \qquad D_\mu g^{-1} = \partial_\mu g^{-1} + a_\mu g^{-1} = -k_\mu g^{-1}. \qquad (2.20)$$

Therefore, for the purpose of enumerating a complete set of operators, one can assume that the covariant derivatives never act on $g$ or $g^{-1}$. Then, $g$ and $g^{-1}$ must appear only in the combination $g^{-1}g = 1$, since the other fields are already invariant under the global $G$ symmetry. This completes the proof.

Now our task boils down to classifying local operators that consist only of the following symbols:

$$D_\mu, \qquad k_\mu^a, \quad a \in \{1, \ldots, \dim \mathfrak{k}\}, \qquad (2.21)$$

where we have now made the Lie algebra index of $k$ explicit. Acting with the covariant derivatives on $k$, one obtains the basic building blocks, which we shall call "letters". Note that acting with $D_\mu$ on $k_\nu$ via (2.9) keeps us within $\mathfrak{k}$ because $[\mathfrak{h}, \mathfrak{k}] \subset \mathfrak{k}$. Examples of such letters are

$$(D_+ k_+)^a, \quad (D_+ D_- k_+)^a, \quad (D_+ D_- D_+ k_-)^a \qquad \text{etc.} \qquad (2.22)$$

We should however keep in mind that not all the letters are independent. Firstly, owing to the relation (2.15), one can effectively treat the covariant derivatives as mutually commuting objects; the non-commuting parts of the covariant derivatives can be expressed in terms of two $k^a$'s using (2.15). Thus we can reduce the set of letters to

$$(D_+)^n (D_-)^m k_+, \qquad (D_-)^n (D_+)^m k_-. \qquad (2.23)$$

Secondly, one can replace the operators of the form $D_- k_+$ or $D_+ k_-$ with operators without covariant derivatives using the equation of motion (2.13) and the relation (2.16), which we display here again in lightcone coordinates,

$$\begin{aligned} D_+ k_- + D_- k_+ &= 0, \\ D_+ k_- - D_- k_+ &= -[k_+, k_-]\big|_{\mathfrak{k}}. \end{aligned} \qquad (2.24)$$

Using these two relations, we get explicit expressions for $D_- k_+$ and $D_+ k_-$ in terms of products of $k^a$'s. Using these expressions in (2.23), we conclude that the complete set of letters is given by

$$k_+^{(n)} := (D_+)^n k_+, \qquad k_-^{(n)} := (D_-)^n k_-. \qquad (2.25)$$

In other words, we only need to consider letters with all plus or all minus indices.

As the final step, we need to impose invariance under the $H$ gauge transformations. (Recall that all the letters are already invariant under the global $G$-symmetry.) For this purpose, we note that because $[\mathfrak{h}, \mathfrak{k}] \subset \mathfrak{k}$, the vector space $\mathfrak{k}$ forms a representation $r$ of $\mathfrak{h}$. The representation $r$ is, in general, not an irrep and we can decompose $r = \oplus_i r_i$, where $r_i$'s are irreps of $H$. For instance, in the case of $O(N)/O(N-1)$ coset, the index $a$ in $k_\mu^a(x)$ can take $N-1$ possible values, and $k$ transforms in the vector representation of $O(N-1)$. In the case of $SU(N+1)/U(N)$ coset, the index $a$ in $k_\mu^a(x)$ can take $2N$ possible values, and $k$ transforms in the $N \oplus \overline{N}$ of $U(N)$. Thus, in general we can write

$$k_\mu = \sum_i [k_\mu]_{r_i}. \qquad (2.26)$$

The covariant derivatives do not change the representations, and so the letters

$$[k_+^{(n)}]_{r_i} := (D_+)^n [k_+]_{r_i}, \qquad [k_-^{(n)}]_{r_i} := (D_-)^n [k_-]_{r_i} \qquad (2.27)$$

also transform in the representation $r_i$. Finally, we need to solve the group-theoretic problem of constructing $H$-invariant objects out of products of the letters in (2.27). We do this in the next section by constructing a generating function for $H$-invariant operators via Haar integration over the group $H$.

# 3 An index for quantum integrability

In this section we introduce an algorithmic approach to diagnose the fate of classically-conserved currents at the quantum level [1]. Our computational techniques are inspired by [25] and [26]. We will first work in the UV (which enjoys conformal invariance) to construct a generating function and to define the index. At the end, we will explain why the index is invariant in the regime of conformal perturbation theory around the UV fixed point.

## 3.1 Generating function for local operators

At the UV fixed point, the equations of motion (2.13) become linear and we have conformal symmetry. This allows us to organize operators by their conformal dimension and spin,

$$Z(q,x) := \sum_{\text{inv}\,\mathcal{O}} q^{\Delta_{\mathcal{O}}} x^{J_{\mathcal{O}}}, \tag{3.1}$$

where we only include operators that are $H$-invariant. As usual, $\Delta_{\mathcal{O}}$ and $J_{\mathcal{O}}$ denote the dimension and the spin of the operator $\mathcal{O}$, respectively.

As discussed in Section 2.2, the complete set of single-letter operators is given by $(D_+)^n k_+$ and $(D_-)^n k_-$ (see (2.25)). This leads to the following generating function for single-letter operators:

$$\widehat{f}(q,x) := \sum_{n=0}^{\infty} q^{n+1}\left(x^{n+1} + x^{-(n+1)}\right) = \frac{xq}{1-xq} + \frac{x^{-1}q}{1-x^{-1}q}. \tag{3.2}$$

In $\widehat{f}$, we only kept track of the scaling dimension and spin, but we also need to keep track of the quantum numbers under $H$ transformations. For this purpose, we introduce the fugacities $y$, which is a vector of length equal to the number of Cartan generators of $H$. We decompose the current $k_\mu^a$ into irreducible representations of $H$ as in (2.26), and multiply by the character for each representation $\chi_{r_i}(y)$. This leads to the following formula for the single-letter generating function $f(q,x,y)$:

$$f(q,x,y) := \widehat{f}(q,x)\,\chi_r(y) = \widehat{f}(q,x)\left(\sum_i \chi_{r_i}(y)\right). \tag{3.3}$$

The next step is to express the multi-letter generating function in terms of the single-letter generating function. To see how the computation goes, let us consider one particular single-letter operator with definite dimension $\Delta$, spin $J$ and charge vector $R$ under the Cartan generators. Such an operator contributes a monomial to the generating function

$$f^{(\Delta,J,R)}(q,x,y) = q^\Delta x^J y^R. \tag{3.4}$$

Here, $y^R$ is a shorthand for $\prod_{i=1}^{\text{rank}\,\mathfrak{h}} y_i^{R_i}$. If we construct multi-letter operators using only this operator, the partition function would read

$$\begin{aligned} Z^{(\Delta,J,R)}(q,x,y) &= 1 + q^\Delta x^J y^R + (q^\Delta x^J y^R)^2 + \cdots \\ &= \exp\left[-\log(1 - q^\Delta x^J y^R)\right] \\ &= \exp\left[\sum_{m=1}^{\infty} \frac{1}{m} f^{(\Delta,J,R)}(q^m, x^m, y^m)\right]. \end{aligned} \tag{3.5}$$

In reality, there are infinitely many single-letter operators and the multi-letter partition function would be given by a product of the factor (3.5) corresponding to each single-letter operator. This leads to the following expression for the multi-letter partition function:

$$Z(q,x,y) = \exp\left[\sum_{m=1}^{\infty} \frac{1}{m} f(x^m, q^m, y^m)\right], \tag{3.6}$$

with $f(x,q,y)$ given in (3.3). The expression on the right hand side is also known as the plethystic exponential [26]. To obtain the generating function for gauge-invariant operators, we simply need to integrate over the fugacities with the Haar measure on $H$:

$$Z(q,x) = \int d\mu_H(y) \, Z(q,x,y) \, . \tag{3.7}$$

Below we will also encounter cases where we cannot restrict Haar integrals to the Cartan. In such cases, we cannot introduce the fugacities $y$, so we need a general element $h \in H$ in our formulas. In particular, the equations (3.3) and (3.6) are replaced by

$$f(q,x,h) = \hat{f}(x,q) \, \chi_r(h) = \hat{f}(x,q) \left( \sum_i \chi_{r_i}(h) \right) , \tag{3.8}$$

$$Z(q,x,h) = \exp\left[ \sum_{m=1}^{\infty} \frac{1}{m} f(x^m, q^m, h^m) \right] . \tag{3.9}$$

The projection to the gauge-invariant operators can be achieved by integrating $Z(q,x,h)$ against the Haar measure $d\mu_H$, generalizing (3.7):

$$Z(q,x) = \int d\mu_H(h) \, Z(q,x,h) \, . \tag{3.10}$$

Equations (3.2), (3.8), (3.9) and (3.10) are our main results that make the computations of [1] algorithmic.

## 3.2 Discrete symmetries

Now we extend these formulas to include discrete symmetries which are crucial for quantum integrability of certain models. To be concrete, let us consider a sigma model with an internal $\mathbb{Z}_2$ symmetry, whose group elements are given by 1 and $\sigma$, with $\sigma^2 = 1$.

One can take the $\mathbb{Z}_2$ symmetry into account by considering the modified partition function

$$\tilde{Z}(q,x) := \frac{1}{2} \left[ Z(q,x) + Z_\sigma(q,x) \right] , \tag{3.11}$$

with $Z(q,x)$ as before (3.1) and

$$Z_\sigma(q,x) := \sum_{\text{inv } \mathcal{O}} \left[ \sigma \, q^{\Delta_\mathcal{O}} x^{J_\mathcal{O}} \right] . \tag{3.12}$$

In other words, we insert $\frac{1+\sigma}{2}$ in the partition function, which projects to the $\mathbb{Z}_2$-invariant sector. Again, we restrict ourselves to analyze operators that are invariant under the global discrete symmetries, but it is straightforward to generalize to operators in nontrivial representations of the symmetry. The formula for $Z_\sigma$ is a straightforward generalization of (3.9).

$$Z_\sigma(q,x,h) = \exp\left[ \sum_{m=1}^{\infty} \frac{1}{m} \hat{f}(x^m, q^m) \operatorname{tr}_r \left( (\sigma h)^m \right) \right] . \tag{3.13}$$

In general, $\sigma$ maps the representation $r$ to itself, but it can take us between the representations $r_i$. For example, in the case of $O(N)/O(N-1)$, there is only one representation, and $\sigma$ keeps us within this representation. In the case of $SU(N+1)/U(N)$, the fundamental and the anti-fundamental representations get exchanged by $\sigma$.

### 3.3 Index for quantum integrability

The partition function (3.9) is defined at the UV free CFT point of the sigma model, restricted to the $H$-invariant sector. Thus, we can expand the partition function of the UV theory into a sum of characters of the two-dimensional global conformal group,

$$Z(q,x) = \sum_{\Delta,J} c(\Delta,J)\, \chi_{\Delta,J}(q,x), \qquad (3.14)$$

where the non-negative integer $c(\Delta,J)$ counts the number of global primaries with dimension $\Delta$ and spin $J$. As reviewed in appendix A, we have two types of characters: short characters for conserved currents and the more typical long characters for everything else.

In terms of $c(\Delta,J)$, the index (1.6) for the UV CFT can be expressed simply as

$$\mathfrak{I}(J) = c(J,J) - c(J+1,J-1). \qquad (3.15)$$

The first term denotes the number of primary conserved currents of spin $J$ in the UV CFT.[5] The second term counts the number of primary operators with dimension $J+1$ and spin $J-1$. This is precisely the type of operators that can appear as $A$ terms in $\partial_- \mathcal{O}_{J,J}$, and cannot be absorbed into a redefinition of the current. Thus, there exists a quantum conserved current if this number is strictly positive. Indeed, this is just the criterion of [1]. The novelty in our work is that we are choosing to work at the UV fixed point which allows us to exploit conformal symmetry.

To summarize, we can diagnose quantum integrability of a coset model by computing the generating function using the formulas (3.2), (3.8)-(3.10), reading off the expansion coefficients (3.14) to construct the index (3.15), and checking

$$\mathfrak{I}(J) > 0 \quad \Longrightarrow \quad \text{There exists a quantum conserved current of spin } J. \qquad (3.16)$$

Further, if $\mathfrak{I}(J) > 0$, the number of quantum conserved currents is at least $\mathfrak{I}(J)$. In appendix A, we discuss an "inversion formula" which allows us to compute $\mathfrak{I}(J)$ as an integral transform of $Z(q,x)$. In practice, for low spin operators, it is often easier to explicitly series expand the partition function $Z(q,x)$ and read off the coefficients $c(\Delta,J)$.[6]

### 3.4 Invariance of the index under conformal perturbation theory

We now comment on an important feature of the index, which is its invariance under conformal perturbation theory around the UV fixed point. When we move away from the UV fixed point, some spin-$J$ conserved current $\mathcal{O}_{J,J}$ can cease to be conserved. This is because the conformal multiplet of $\mathcal{O}_{J,J}$ can combine with a multiplet whose primary $\mathcal{O}_{J+1,J-1}$ has dimension $J+1$ and spin $J-1$. In the process, the conformal multiplet of $\mathcal{O}_{J,J}$ becomes a long multiplet that satisfies the relation $\partial_- \mathcal{O}_{J,J} = \mathcal{O}_{J+1,J-1}$. When this happens, the first term in (1.6) reduces by one. At the same time, the third term in (1.6) also increases by one since now the operator $\mathcal{O}_{J+1,J-1}$ is a total divergence. As a result, the difference $\mathfrak{I}(J)$ remains invariant.

To see this in a concrete example, let us consider the case of the $\mathbb{CP}^{N-1}$ model, which will be discussed in more detail in Section 4.2 below. In computing $\mathfrak{I}(4)$ for this case, we will find that $c(4,4) = c(5,3) = 2$, and so $\mathfrak{I}(4) = 2 - 2 = 0$. Let us compare this to [1]. They find that

---

[5]Note that descendants can also satisfy a conservation law, but being total derivatives they do not give rise to a charge when integrated on a spatial slice.

[6]For practical computations it is also useful to note that $\mathfrak{I}(J) = a(J,J) - a(J-1,J-1) - a(J+1,J-1) + a(J,J-2)$ where $a(\Delta,J)$ is the coefficient of $q^{\Delta}x^J$ in the expansion of $Z(q,x)$. Note also that $a(\Delta,J)$ would be the total number of operators taking into account the full non-linear equations of motion, because (2.25) is the complete set of letters. In particular, $a(J+1,J-1)$ would be the number of $A$ terms in [1].

there is just one candidate conserved operator with $\Delta = J = 4$, namely the operator $(T_{++})^2$. They also find four $A$ operators and three $B$ operators, and thus one primary with $\Delta = 5$ and $J = 3$. Thus, with their way of counting, the index would be $\mathcal{I}(4) = 1 - 1 = 0$. The reason for the discrepancy is the following. In the free limit, one operator with $\Delta = J = 4$ is $(T_{++})^2$, and let us call the other one $\mathcal{O}_{4,4}$. The free equations of motion imply that $\partial_- \mathcal{O}_{4,4} = 0$. What happens as we flow away from the UV is that we get a modified relation $\partial_- \mathcal{O}_{4,4} = \mathcal{O}_{5,3}$, where $\mathcal{O}_{5,3}$ is one of the primary operators contributing to $c(5,3) = 2$. Thus, we lose one conserved operator because $\mathcal{O}_{4,4}$ is no longer conserved, and we lose one $A$ term because $\mathcal{O}_{5,3}$ is now a total divergence. As a result, the index remains invariant.

The above argument is valid in the regime of conformal perturbation theory, where we can grade the local operators by their scaling dimensions at the UV fixed point. There is a potential subtlety related to nonperturbative corrections. The coset sigma models discussed in this paper are asymptotically free, but acquire a mass gap nonperturbatively in the infrared. In the presence of such a mass gap, one can write $A$ terms for $\partial_- \mathcal{J}_{+\dots+}$ with dimension less than $J + 1$. The sufficiency condition of [1] can in principle be violated by this nonperturbative effect, but we are not aware of any example where this happens.

# 4 Examples

## 4.1 $\mathbb{CP}^1$ model

We now apply the strategy above to the $\mathbb{CP}^1$ model with a general $\theta$ angle. The coset for the $\mathbb{CP}^1$ sigma model is $\frac{SU(2)}{U(1)}$. The $\mathbb{CP}^1$ sigma model is integrable at $\theta = 0$ [2] and at $\theta = \pi$ [27], and the global symmetry at these two points is $O(3) = SO(3) \rtimes \mathbb{Z}_2$. The $\mathbb{CP}^1$ model is not expected to be integrable for other values of the $\theta$, where the global symmetry is simply $SO(3)$.

Let us first compute the index without imposing the charge conjugation symmetry, corresponding to the $\mathbb{CP}^1$ sigma model with a generic $\theta$ angle. The coset degrees of freedom consist of the charge $+1$ representation and the charge $-1$ representation of the $U(1)$ quotient group. The $U(1)$ character is simply $\text{tr}(h) = y + y^{-1}$, where $y = e^{i\phi}$ is the $U(1)$ fugacity. Hence,

$$\text{tr}(h^m) = y^m + y^{-m} \,. \tag{4.1}$$

The multi-letter partition function $Z(q, x, y)$ is constructed following (3.2), (3.3) and (3.6). We project to $U(1)$ invariant operators using (3.7), which in this case becomes

$$Z(q, x) = \oint \frac{dy}{2\pi i y} Z(q, x, y) \,. \tag{4.2}$$

We get the following result for the indices:

$$\mathcal{I}(4) = 0 \,, \quad \mathcal{I}(6) = -1 \,, \quad \mathcal{I}(8) = -5 \,, \quad \mathcal{I}(10) = -15 \,, \quad \mathcal{I}(12) = -33 \,, \cdots \tag{4.3}$$

Recall that $\mathcal{I}(J) > 0$ is a sufficient condition for the existence of quantum conserved spin $J$ currents. Hence without imposing charge conjugation symmetry, our analysis does not predict quantum conserved currents for the $\mathbb{CP}^1$ model, consistent with the expectation that the $\mathbb{CP}^1$ model is not integrable at a generic $\theta$ angle.

Next, we compute the index for the $\mathbb{CP}^1$ sigma model at $\theta = 0, \pi$, where there is a $\mathbb{Z}_2$ charge conjugation symmetry and the model is known to be integrable. The $\mathbb{Z}_2$ charge conjugation symmetry maps a charge $+1$ state to a charge $-1$ state, and extends the quotient group from $U(1)$ to $O(2)$. The $k_\mu^a$ form a two-dimensional representation of the $O(2)$ group in which

the group element can be expressed as

$$h = \begin{pmatrix} \cos\phi & \sin\phi \\ -\sin\phi & \cos\phi \end{pmatrix}, \quad \sigma = \begin{pmatrix} 1 & 0 \\ 0 & -1 \end{pmatrix}, \tag{4.4}$$

with $\phi \in [0, 2\pi)$. Using this matrix representation, the trace $\mathrm{tr}((\sigma h)^m)$ can be computed straightforwardly and we get

$$\mathrm{tr}((\sigma h)^m) = 1 + (-1)^m. \tag{4.5}$$

Now we can compute the full partition function for the $\mathbb{CP}^1$ sigma model with charge conjugation symmetry

$$\tilde{Z}(q, x) = \oint \frac{dy}{2\pi i y} \frac{1}{2} \left[ Z(q, x, y) + Z_\sigma(q, x, y) \right], \tag{4.6}$$

with $Z_\sigma$ computed via (3.13) using (4.5).

Using this new partition function, we get the following indices:

$$\mathcal{I}(4) = 1, \quad \mathcal{I}(6) = 1, \quad \mathcal{I}(8) = 0, \quad \mathcal{I}(10) = -4, \quad \mathcal{I}(12) = -11 \cdots \tag{4.7}$$

Thus, there exist quantum conserved currents of spin-4 and spin-6, making the model integrable after incorporating discrete symmetry. The existence of the spin-4 current was shown in the original work of [1], and we further established that there is a spin-6 current. Our analysis does not predict conserved currents of even higher spin.[7] We will see in Section 4.3 that this spin-6 quantum conserved current also exists in all the $O(N)$ models.

## 4.2 $\mathbb{CP}^{N-1}$ model

The $\mathbb{CP}^{N-1}$ model is the sigma model with target space $\frac{SU(N)}{U(N-1)}$. The index $a$ in $k_\mu^a$ transforms in a direct sum of the fundamental and the anti-fundamental representations of $U(N-1)$. The characters for these representations are given by

$$\chi_\square(y_1, \ldots, y_{N-1}) = \sum_k y_k, \qquad \chi_{\bar{\square}}(y_1, \ldots, y_{N-1}) = \sum_k y_k^{-1}. \tag{4.8}$$

The integration measure is given by

$$\int d\mu(y) = \frac{1}{(N-1)!} \left( \prod_{k=1}^{N-1} \oint \frac{dy_k}{2\pi i y_k} \right) \prod_{i<j} (y_i - y_j)(y_i^{-1} - y_j^{-1}). \tag{4.9}$$

Computing the index using these formulae we obtain

$$\mathcal{I}(4) = -2, \quad \mathcal{I}(6) = -6, \tag{4.10}$$

independent of $N$. Since all these numbers are negative, it is unlikely that there are conserved higher-spin currents at the quantum level. Let us see if imposing charge conjugation symmetry can help.

The $U(N-1)$ group element and charge conjugation matrix $\sigma$ in the representation $r = \square \oplus \bar{\square}$ are given by

$$r(h) = \begin{pmatrix} h & 0 \\ 0 & h^* \end{pmatrix}, \quad \sigma = \begin{pmatrix} 0 & I_{N-1} \\ I_{N-1} & 0 \end{pmatrix}, \tag{4.11}$$

---

[7]Incidentally, the indices for the charge conjugation odd currents are all negative.

where $h \in U(N-1)$. The traces $\mathrm{tr}[(\sigma h)^m]$ needed in (3.13) vanish for odd $m$ and for even $m$ reduce to $2\,\mathrm{tr}[(hh^*)^{\frac{m}{2}}]$. We now compute Haar integrals analytically over $h$ using the so-called Weingarten functions for the unitary group. The first two examples are

$$\int dU\, U_{ij}U^*_{i'j'} = \frac{1}{d}\,\delta_{ii'}\delta_{jj'}\,, \tag{4.12}$$

$$\int dU\, U_{i_1j_1}U_{i_2j_2}U^*_{i'_1j'_1}U^*_{i'_2j'_2} = \frac{\delta_{i_1i'_1}\delta_{i_2i'_2}\delta_{j_1j'_1}\delta_{j_2j'_2} + \delta_{i_1i'_2}\delta_{i_2i'_1}\delta_{j_1j'_2}\delta_{j_2j'_1}}{d^2-1}$$
$$-\frac{\delta_{i_1i'_1}\delta_{i_2i'_2}\delta_{j_1j'_2}\delta_{j_2j'_1} + \delta_{i_1i'_2}\delta_{i_2i'_1}\delta_{j_1j'_1}\delta_{j_2j'_2}}{d(d^2-1)}\,. \tag{4.13}$$

Here $dU$ is the Haar measure on $U(d)$ normalized such that $\int dU = 1$.[8] The result of the index computation is that

$$\mathcal{I}(4) = 0\,, \quad \mathcal{I}(6) = -1\,, \tag{4.14}$$

independent of $N$. The discrete symmetry increases the indices, but they are still not positive, and so the classically-conserved currents may not survive quantum-mechanically. This is consistent with the fact that the $\mathbb{CP}^{N-1}$ models with $N > 2$ are not expected to be integrable [36, 37].

## 4.3 $O(N)$ model

The $O(N)$ model can be viewed as the sigma model with target space $\frac{SO(N)}{SO(N-1)}$. In other words the target space is the sphere $S^{N-1}$. For simplicity, we assume that $N-1$ is even. The index $a$ in the current $k^a_\mu$ transforms under the vector representation of $SO(N-1)$, and its character is given by

$$\chi(y) = \sum_{i=1}^{(N-1)/2} (y_i + y_i^{-1})\,. \tag{4.15}$$

The measure factor for integrating over the Cartan is given by

$$d\mu(y) = \prod_i \frac{dy_i}{2\pi i\, y_i} \prod_{i<j}(1 - y_iy_j)(1 - \frac{y_i}{y_j})\,. \tag{4.16}$$

Using these formulas, together with (3.2), (3.3), (3.6) and (3.7), one can compute $\mathcal{I}(J)$. The results for small $N$ are summarized in Table 1, and for the spin-4 case agree with the findings in [1]. Since $\mathcal{I}(4) > 0$ for all values of $N$ except $N = 3$, this shows quantum integrability for $N > 3$. For $N = 3$, which is the same as the $\mathbb{CP}^1$ model, we need to take into account discrete symmetries, as we also saw in Section 4.1.

So we now proceed to impose the $\mathbb{Z}_2$ charge conjugation symmetry which extends the quotient group from $SO(N-1)$ to $O(N-1)$. Since the charge conjugation $\sigma = \mathrm{diag}(1, 1, \ldots, 1, -1)$ maps the vector representation of $SO(N-1)$ to itself, the computation of the modified partition function (3.13) boils down to computing $\mathrm{tr}[(\sigma h)^m]$ in the vector representation and integrating the plethystic exponential over $SO(N-1)$. This integral cannot be reduced to an integral over the Cartan since charge conjugation does not commute with generic group elements of $SO(N-1)$.[9] Nevertheless, as shown in Appendix C of [25], one can still simplify the integral

---

[8]See [35] for a Mathematica package that computes Weingarten integrals symbolically.
[9]Recall that $O(N-1) = SO(N-1) \rtimes \mathbb{Z}_2$.

Table 1: The first few indices for the $O(N)$ sigma model $\frac{SO(N)}{SO(N-1)}$. On the left are indices without imposing any discrete symmetry, and on the right are indices when we impose the charge conjugation symmetry. We take $N$ to be odd for simplicity. The case with $N = 3$ is the same as the $\mathbb{CP}^1$ case considered in Section 4.1. Thus our analysis confirms the presence of a spin-4 conserved current and we predicts a new spin-6 conserved current at the quantum level.

|         | $\mathcal{I}(4)$ | $\mathcal{I}(6)$ | $\mathcal{I}(8)$ |
|---------|------|------|------|
| $N = 3$ | 0 | $-1$ | $-5$ |
| $N = 5$ | 1 | 0 | $-1$ |
| $N = 7$ | 1 | 1 | 0 |
| $N = 9$ | 1 | 1 | 0 |

|         | $\mathcal{I}(4)$ | $\mathcal{I}(6)$ | $\mathcal{I}(8)$ |
|---------|------|------|------|
| $N = 3$ | 1 | 1 | 0 |
| $N = 5$ | 1 | 1 | 0 |
| $N = 7$ | 1 | 1 | 0 |
| $N = 9$ | 1 | 1 | 0 |

into multiple abelian integrals. Their analysis is based on the fact that, for any $h \in SO(N-1)$, $\sigma h$ can be brought to the following block-diagonal matrix by conjugation,

$$\sigma h \mapsto \begin{pmatrix} R_1 & \cdots & 0 & 0 \\ \vdots & \ddots & \vdots & \vdots \\ 0 & \cdots & R_{\frac{N-3}{2}} & 0 \\ 0 & \cdots & 0 & J \end{pmatrix}, \tag{4.17}$$

with

$$R_k = \begin{pmatrix} \cos\theta_k & \sin\theta_k \\ -\sin\theta_k & \cos\theta_k \end{pmatrix}, \qquad J = \begin{pmatrix} 1 & 0 \\ 0 & -1 \end{pmatrix}. \tag{4.18}$$

We just state the outcome, referring to [25] for details: The modified partition function $Z_\sigma$ can be computed by replacing the character and the measure with

$$\tilde{\chi}(y) = \tilde{y}_+ + \tilde{y}_- + \sum_{i=1}^{(N-3)/2} (y_i + y_i^{-1}), \tag{4.19}$$

$$d\tilde{\mu}(y) = \frac{d\tilde{y}_+}{2\pi i(\tilde{y}_+ - 1)} \frac{d\tilde{y}_-}{2\pi i(\tilde{y}_- + 1)} \prod_i \frac{dy_i(1 - y_i^2)}{2\pi i\, y_i} \prod_{i<j}(1 - y_iy_j)(1 - \frac{y_i}{y_j}), \tag{4.20}$$

where the integration contours for $\tilde{y}_\pm$ are around $\pm 1$ respectively. Using these expressions, we computed $\mathcal{I}(J)$ for small odd $N$ (including $N = 3$) and found that

$$\mathcal{I}(4) = 1, \qquad \mathcal{I}(6) = 1, \qquad \mathcal{I}(8) = 0, \tag{4.21}$$

independent of $N$. Thus our analysis is consistent with quantum integrability of the $O(N)$ model with $\mathbb{Z}_2$ symmetry. In addition to the spin-4 conserved current established in [1], we have predicted a spin-6 conserved current at the quantum level.[10]

## 4.4 Flag sigma models $\frac{U(N)}{U(1)^N}$

As one last example, we compute the index for the flag sigma model $\frac{U(N)}{U(1)^N}$, which has been studied recently in [30–32]. This is also an example where the coset is not symmetric (for

---

[10]Our index analysis does not predict the existence of the conserved currents with spin > 8 although it is likely that there exist infinitely many higher-spin conserved currents given that the model is integrable. Note also that in [38] it was claimed that there exists a quantum conserved current for each even spin. This is incorrect as [38] overcounts operators that are total derivatives, because they include operators which vanish owing to the equation of motion.

Table 2: The first few indices for the flag sigma model $\frac{U(N)}{U(1)^N}$. On the left are indices without imposing any discrete symmetry, and on the right are indices while imposing the $S_N \times \mathbb{Z}_2$ symmetry. The indices are all negative, which means that our counting analysis does not predict higher-spin quantum conserved currents.

|         | $\mathcal{I}(4)$ | $\mathcal{I}(6)$ | $\mathcal{I}(8)$ |
|---------|--------|---------|----------|
| $N = 3$ | $-47$   | $-262$   | $-1263$   |
| $N = 4$ | $-371$  | $-3834$  | $-32235$  |
| $N = 5$ | $-1605$ | $-27794$ | $-379760$ |

|         | $\mathcal{I}(4)$ | $\mathcal{I}(6)$ | $\mathcal{I}(8)$ |
|---------|--------|--------|--------|
| $N = 3$ | $-4$   | $-20$   | $-105$  |
| $N = 4$ | $-7$   | $-79$   | $-682$  |
| $N = 5$ | $-10$  | $-139$  | $-1722$ |

$N > 2$). Note that the $N = 2$ flag sigma model is the $O(3)$ or the $\mathbb{CP}^1$ model. The flag sigma model has a $N(N-1)$-dimensional parameter space preserving the $PSU(N)$ global symmetry. Over special loci on the parameter space, the model has enhanced discrete symmetries and 't Hooft anomalies. It has been argued that over certain special loci on the moduli space the model is gapless in the IR and is described by the $SU(N)_1$ WZW model [30–32].

The index $a$ in $k_\mu^a$ takes $(N^2 - N)$ possible values corresponding to the roots of $SU(N)$. The charge of $(k_\mu)^{ij}$ (with $i, j = 1, \dots, N$, $i \neq j$) under the $n$-th $U(1)$ factor in $U(1)^N$ is $\delta_{i,n} - \delta_{j,n}$. The required $U(1)^N$ character is

$$\chi(y) = \sum_{\substack{i,j=1 \\ i \neq j}}^{N} y_i y_j^{-1}. \tag{4.22}$$

We first computed the indices $\mathcal{I}(J)$ without imposing any discrete symmetry, and the results are given on the left in Table 2. All the indices are negative. Hence our analysis does not predict higher-spin quantum conserved currents in the flag sigma model at a generic point in the parameter space.

Next, we compute the index at the "origin" of the parameter space, where the enhanced discrete symmetry is $S_N \times \mathbb{Z}_2$, with $S_N$ the permutation group on $N$ elements. A permutation $\sigma \in S_N$ acts on the current $(k_\mu)^{ij}$ via $(k_\mu)^{ij} \to (k_\mu)^{\sigma(i)\sigma(j)}$, while the $\mathbb{Z}_2$ acts as $(k_\mu)^{ij} \to (k_\mu)^{ji}$. We define a $N(N-1) \times N(N-1)$ diagonal matrix

$$h = \text{diag}\left(y_1 y_2^{-1}, y_1 y_3^{-1}, \cdots, y_{N-1} y_N^{-1}, y_1^{-1} y_2, y_1^{-1} y_3, \cdots, y_{N-1}^{-1} y_N\right), \tag{4.23}$$

whose trace is given in (4.22). For each element $\sigma$ of $S_N \times \mathbb{Z}_2$, we write down its matrix representation acting on $\mathfrak{k}$, and compute $\text{tr}[(\sigma h)^m]$.[11] The partition function for the $S_N \times \mathbb{Z}_2$ invariant operators is then

$$\frac{1}{2N!} \prod_{i=1}^{N} \oint \frac{dy_i}{2\pi i y_i} \sum_{\sigma \in S_N \times \mathbb{Z}_2} Z_\sigma(q, x, y_i), \tag{4.24}$$

where $Z_\sigma$ is as in (3.13). We find that all the indices are negative. See Table 2. If we impose a smaller subgroup of $S_N \times \mathbb{Z}_2$, the indices are even more negative. Thus, we conclude that our analysis does not predict higher-spin quantum conserved currents for the flag sigma model anywhere on the parameter space. This in particular suggests that the classical integrability of the flag sigma model on $\frac{U(3)}{U(1)^3}$ found in [39] is likely to be broken at the quantum level.

---

[11]For example, the charge conjugation $\mathbb{Z}_2$ is realized as $\sigma = \begin{pmatrix} 0 & I_{\frac{N(N-1)}{2}} \\ I_{\frac{N(N-1)}{2}} & 0 \end{pmatrix}$.

# 5 Conclusions and future directions

In this paper, we systematized the analysis of Goldschmidt and Witten [1] by exploiting the conformal symmetry of coset models in the UV. We introduced the index $\mathcal{I}(J)$, eqns. (1.6) and (3.15), whose positivity for spin $J > 2$ gives a sufficient condition for quantum integrability. We also discussed the invariance of the index under conformal perturbation theory around the UV fixed point. We applied our formalism in several examples and found the following results:

1. The $\mathbb{CP}^1$ model (Section 4.1) is integrable at $\theta = 0$ and $\theta = \pi$, where there is a $\mathbb{Z}_2$ charge conjugation symmetry, since $\mathcal{I}(4)$ and $\mathcal{I}(6)$ are positive. On the other hand, without imposing the extra $\mathbb{Z}_2$ symmetry, the indices are all non-positive, consistent with the standard lore that the $\mathbb{CP}^1$ model is not integrable away from $\theta = 0, \pi$.

2. The indices for the $\mathbb{CP}^{N-1}$ model (Section 4.2) with $N \geqslant 3$ are all non-positive, consistent with the fact that they are not quantum integrable [36, 37].

3. For the $O(N)$ model (Section 4.3 and Table 1), we found that $\mathcal{I}(4) = \mathcal{I}(6) = 1$ (with a $\mathbb{Z}_2$ symmetry), thereby establishing the existence of a spin-6 conserved current in addition to the well-known spin-4 conserved current.

The examples above are symmetric cosets which are known to be classically integrable, but our analysis is also applicable to more general cosets. As an example, we studied the $\frac{U(N)}{U(1)^N}$ flag sigma models and found that

4. The indices for the flag sigma models (Section 4.4 and Table 2) are all negative even after imposing the maximum amount of discrete symmetry. Thus it is unlikely that these models are integrable.

We now remark on some avenues for future work.

As demonstrated in the example of the $\mathbb{CP}^1$ model, discrete symmetry plays an important role for quantum integrability. However, our analysis is not sensitive to potential 't Hooft anomalies, which can have consequences for integrable flows. For example, while the $\mathbb{CP}^1$ model has $O(3) = SO(3) \rtimes \mathbb{Z}_2$ global symmetry both at $\theta = 0$ and $\theta = \pi$, the 't Hooft anomalies are different at these two points. At $\theta = 0$, there is no anomaly, while at $\theta = \pi$, there is a mixed anomaly between $SO(3)$ and the $\mathbb{Z}_2$ charge conjugation symmetry [32, 40, 41]. Relatedly, the IR phases at $\theta = 0$ and at $\theta = \pi$ are different. At $\theta = 0$, the IR is trivially gapped, while at $\theta = \pi$, the IR phase is gapless and is described by the $SU(2)_1$ WZW model which captures the mixed anomaly. One potential avenue to incorporate the information from 't Hooft anomalies into our index would be to interpret it as a torus partition function (possibly with symmetry lines inserted), whose modular transformation generally depends on the 't Hooft anomaly (see, for example, [42–44]).

Our analysis can be extended to supersymmetric theories and theories with fermions. For instance, it is known that the $\mathbb{CP}^{N-1}$ models can be made quantum integrable by coupling them to fermions [37, 45], and it would be interesting to see if the same is true for the flag sigma models.

Using the idea developed in this paper, one can also analyze "fine-tuned" quantum integrability: Some theories [45] can be made quantum integrable after tuning the coefficients for marginal operators. This can be diagnosed by computing a "refined" index

$$\mathcal{I}_r(J) := \mathcal{I}(J) + c(2,0). \tag{5.1}$$

Here $c(2,0)$ is the number of marginal primary operators in the UV. Unlike $\mathcal{I}(J)$ discussed in the paper, $\mathcal{I}_r(J) > 0$ is not a sufficient condition for quantum integrability, but having $\mathcal{I}_r(J) > 0$

will make it more likely for quantum integrability to be achieved at some point in parameter space.

It should also be possible to extend our analysis to deformations of sigma models which partially break the global $G$ symmetry. One famous example is the sausage model [46] (see [47] for a recent discussion), which is an integrable deformation of the $O(3)$ model. The integrability of such models can be analyzed by generalizing our computation to operators which are not invariant under $G$.

Finally, it would be interesting if one can generalize our analysis to superstring sigma models and find new integrable backgrounds.[12]

## Acknowledgments

We would like to thank Z. Komargodski, K. Ohmori, P. Orland, N. Seiberg, E. Witten, M. Yamazaki, and X. Yin for useful conversations. We thank B. Basso and K. Zarembo for comments on a draft. SK is supported by DOE grant number DE-SC0009988. RM is supported by US Department of Energy grant No. DE-SC0016244. The work of SHS is supported by the National Science Foundation grant PHY-1606531 and by the Roger Dashen Membership. This work benefited from the 2019 Pollica summer workshop, which was supported in part by the Simons Foundation (Simons Collaboration on the Non-Perturbative Bootstrap) and in part by the INFN. SHS is grateful for the hospitality of the Physics Department of National Taiwan University during the completion of this work.

## A    Inversion formula for $\mathcal{I}(J)$

Let us first discuss the characters for the global conformal group $SL(2, \mathbb{C})$. For long representations, the characters can be computed easily by summing over all possible operators in the module, namely all operators of the form $(\partial_+)^n (\partial_-)^m \mathcal{O}_{\Delta, J}$. This leads to

$$\chi^l_{\Delta, J}(q, x) = q^\Delta x^J \sum_{n,m} q^{n+m} x^{n-m} = \frac{q^\Delta x^J}{(1-qx)(1-qx^{-1})}. \tag{A.1}$$

On the other hand, the characters for the short representations are given by linear combinations of (A.1). For instance, the conserved current with spin $J$ (and dimension $J$) has the following character

$$\chi^s_{J, J}(q, x) = \chi^l_{J, J}(q, x) - \chi^l_{J+1, J-1}(q, x), \tag{A.2}$$

where the subtraction $-\chi^l_{J+1, J-1}$ amounts to eliminating the null state $\partial_- \mathcal{O}_{J, J}$.

If the characters formed an orthogonal basis, one would be able to extract the coefficients $c(\Delta, J)$ in (3.14) by writing an "inversion formula". The problem is that, typically, such an inversion formula exists only for the principal series representations, but not for physical representations. Fortunately, all the representations relevant for us have integer conformal dimensions and there exists an orthogonality relation which works for such operators:

$$\int d\mu_{x,q} \, \chi^l_{\Delta, J}(q^{-1}, x^{-1}) \, \chi^l_{\Delta', J'}(q, x) = \delta_{\Delta, \Delta'} \delta_{J, J'}, \tag{A.3}$$

---

[12]The classification of classically integrable backgrounds was performed in [48]. See also [49–51] for discussions on quantum integrability of string backgrounds based on factorized scattering.

where the measure is given by

$$\int d\mu_{x,q} = \oint \frac{dq}{2\pi i q} \oint \frac{dx}{2\pi i x} (1-qx)(1-q^{-1}x)(1-q^{-1}x^{-1})(1-qx^{-1}). \qquad \text{(A.4)}$$

One can easily verify that the characters (A.1) are orthogonal under this measure. Therefore, we can give a formula for the index (3.15)

$$\begin{aligned}
\mathcal{I}(J) &= \int d\mu_{x,q} \, \chi_{J+1,J-1}(x^{-1},q^{-1}) \, Z(q,x) \\
&= \oint \frac{dq}{2\pi i q^J} \oint \frac{dx}{2\pi i x^J} (q^{-1}-x)(q^{-1}-x^{-1}) \, Z(q,x).
\end{aligned} \qquad \text{(A.5)}$$

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
