# Peer review of "An Index for Quantum Integrability"

_SciPost Physics, doi:SciPost Phys. 7, 065 (2019)_

## Round 1 · Referee Report · Anonymous (Referee 1) · 2019-9-13

Strengths

1) Systematic construction of an index giving a lower bound on the number of quantum conserved currents of some classically integrable sigma modelsa 2) Article technically sound, interesting and well written

Weaknesses

1) The limit of the method is that it does not allow to conclude that a model is not quantum integrable.

Report

The authors give a systematic construction of an index which is a lower bound on the number of quantum conserved currents of a given spin in some classically integrable sigma models. This is a generalization of a method originally developed by Y. Goldschmidt and E. Witten. Whereas if the index is strictly positive one can conclude that there exists quantum conserved currents, no definite conclusion can be reached when the index is negative or zero. However, the authors check that in many cases where the models are believed not to be quantum integrable, all indices that they compute are negative or zero

---

## Round 1 · Referee Report · Anonymous (Referee 2) · 2019-11-6

Report

There is no space here to go into all details of the rather large topic of interest for the present paper, but we can summarize by saying that it proposes an algorithmic procedure to implementing the sufficient criterion of [1] for counting higher spin local (conserved) currents in coset sigma models at quantum level by starting from classical conservation relations (i.e. evaluating the possibility of quantum anomaly). Then Parke’s theorem [5] (but see also [4]) would ensure integrability, at least under the aspect of the factorization of the S matrix.

The calculations are based on the methods of [24] and [25] and, analogously, give rise to a plethystic exponential formula to be integrated on the Haar measure. As a novelty, the paper counts the ‘integrability index’ at the 2D CFT ultraviolet point and then it proves that it is perturbatively invariant. Besides, the non-perturbative possibility of violating the criterion of [1] by constructing anomalous A terms, with the dynamically generated mass (gap) (and lower dimension operators), is very well illustrated and considered (end of section 3), and so far we cannot say we have seen this mechanism at work (neither in Zamolodchikov’s criterion), though it would be very, very interesting. In the end (section 4), the paper presents also an useful set of examples where the algorithm works efficiently and which clarify the same.

In general the paper is very clear, consequential, solid and well written.

Finally, for all these reasons I would accept the paper for publication.

A typo: at page 6, “…emphasize that fact that we….” should be “…emphasize the fact that we….”

---

## Editorial Decision

published